# REGULARIZED PHYSICS-INFORMED NEURAL NETWORKS FOR NUMERICAL SOLVING OF INVERSE PROBLEMS IN DYNAMICAL SYSTEMS

**Shcherbina Yakov, Krivorotko Olga & Neverov Andrew** *
Scientific center for information technologies and artificial intelligence
Sirius University
Sirius, Sochi, Krasnodarsliy kray, Russia
yascher03@bk.ru, krivorotko.oi@talantiuspeh.ru

## ABSTRACT

Physics-Informed Neural Networks (PINNs) have emerged as a powerful tool for solving forward and inverse problems governed by differential equations by embedding physical laws directly into the learning process, see Raissi et al. (2019) for more information. In epidemiological modeling, PINNs offer a promising framework for reconstructing hidden state trajectories and identifying key epidemiological parameters from sparse and noisy data. However, training PINNs for inverse problems remains challenging due to strong imbalance between loss components associated with governing equations, initial conditions, and observational data. In this work, we propose a regularized PINN approach for inverse epidemic modeling based on the classical SIR model. The method incorporates normalization of time and state variables, weighted loss balancing, and hybrid optimization strategies. Numerical experiments demonstrate improved training stability, enhanced physical consistency, and more accurate parameter identification under limited data availability.

## 1    INTRODUCTION

Mathematical modeling of infectious disease spread plays a crucial role in epidemiology, public health planning, and decision-making processes (Qian et al., 2025). Compartmental models, such as the classical susceptible–infected–recovered (SIR) model, provide a simple yet effective framework for describing the temporal dynamics of epidemics. Despite their conceptual simplicity, these models allow one to capture key mechanisms of disease transmission and recovery through a small number of parameters with clear epidemiological interpretation.

In the SIR model, the dynamics of disease spread are governed by two main parameters: the transmission rate, which characterizes the intensity of contacts leading to new infections, and the recovery rate, which describes the rate at which infected individuals leave the infectious compartment. Accurate estimation of these parameters is essential for reliable forecasting, evaluation of intervention strategies, and assessment of epidemic severity. In particular, the basic reproduction number, defined as the ratio of the transmission and recovery rates, serves as a fundamental indicator of epidemic potential and is widely used in epidemiological analysis.

Traditionally, parameter estimation in compartmental epidemic models is performed by solving an inverse problem based on observed epidemiological data. Classical approaches rely on numerical integration of the governing ordinary differential equations combined with least-squares fitting or maximum likelihood estimation. However, these methods often suffer from several limitations. They require repeated numerical solution of the model, can be sensitive to noise in the data, and may exhibit poor robustness when observations are sparse or incomplete. Moreover, inverse problems in epidemiology are frequently ill-posed: different combinations of parameters can produce similar model outputs, leading to issues of non-identifiability and unstable parameter recovery.

---

*Use footnote for providing further information about author (webpage, alternative address)—*not* for acknowledging funding agencies. Funding acknowledgements go at the end of the paper.

In recent years, physics-informed neural networks (PINNs) have emerged as a promising framework for solving forward and inverse problems governed by differential equations. PINNs incorporate the underlying physical or mathematical laws directly into the training process of a neural network by embedding the governing equations into the loss function. This approach enables the construction of surrogate models that simultaneously approximate the solution of the differential equations and infer unknown parameters from data, without the need for explicit numerical solvers at each optimization step.

The application of PINNs to epidemiological modeling offers several important advantages. First, PINNs naturally enforce the structure of the dynamical system, ensuring consistency with the governing equations even in the presence of noisy or incomplete observations. Second, they allow for a unified treatment of forward and inverse problems within a single learning framework. Third, PINNs can leverage automatic differentiation to accurately compute derivatives, which is particularly beneficial for parameter identification in systems of ordinary differential equations.

Despite these advantages, the use of PINNs for inverse problems in epidemic models raises several open questions. In particular, the identifiability of model parameters strongly depends on the type and amount of available data. In realistic scenarios, epidemiological observations are often limited to a single compartment, such as the number of infected individuals, while other state variables remain unobserved. Under such conditions, the inverse problem may become ill-conditioned, and standard PINN formulations may fail to recover all parameters accurately. Furthermore, strong correlations between parameters can significantly affect the stability of the training process and the interpretability of the results.

The aim of this work is to investigate the applicability of physics-informed neural networks to the solution of forward and inverse problems for the SIR model, with a particular focus on parameter identifiability and sensitivity. We analyze scenarios with different levels of data availability, including cases where full information about all compartments is provided and cases where only partial observations are available. Additionally, we explore a reduced parameterization of the model in terms of the basic reproduction number, which allows us to decrease the dimensionality of the parameter space and improve the robustness of parameter estimation.

## 2 PROBLEM FORMULATION

### 2.1 SIR EPIDEMIC MODEL

We consider the classical SIR model (Kermack & McKendrick, 1927)

$$
\begin{cases}
\frac{dS}{dt} = -\beta SI, \\
\frac{dI}{dt} = \beta SI - \gamma I, \\
\frac{dR}{dt} = \gamma I,
\end{cases}
\tag{1}
$$

where $S(t)$, $I(t)$, and $R(t)$ denote the susceptible, infected, and recovered populations, respectively, and $\beta > 0$, $\gamma > 0$ are unknown epidemiological parameters. The total population size is assumed to be constant and normalized to unity. The model is supplemented with initial conditions

$$
S(0) = S_0, I(0) = I_0, R(0) = R_0,
\tag{2}
$$

satisfying the normalization constraint

$$
S_0 + I_0 + R_0 = 1.
\tag{3}
$$

A fundamental quantity associated with the SIR model is the basic reproduction number

$$
\tilde{R}_0 = \frac{\beta}{\gamma}
\tag{4}
$$

which characterizes the average number of secondary infections caused by a single infected individual in a fully susceptible population. The value of $\tilde{R}_0$ determines the qualitative behavior of the epidemic: if $\tilde{R}_0 > 1$, an outbreak occurs, whereas if $\tilde{R}_0 \leq 1$ the infection dies out.

## 2.2 FORWARD PROBLEM

The forward problem consists in computing the temporal evolution of the state variables $S(t), I(t), R(t)$ satisfying the system of equations (1) for given model parameters $\beta, \gamma$, and prescribed initial conditions (2).

In this work, the forward problem serves two purposes. First, it is used to generate synthetic reference solutions for validating the physics-informed neural network framework. Second, it provides a baseline for assessing the accuracy of the PINN-based approximation of the SIR dynamics.

## 2.3 INVERSE PROBLEM AND PARAMETER IDENTIFICATION

The inverse problem addressed in this study is the identification of unknown model parameters from observational data (5) from system (1). Depending on the scenario, the available data may include observations of all state variables or only a subset of them.

$$
\begin{cases}
S(t_k) = S_k \\
I(t_k) = I_k \qquad , k = 1, ..., K. \\
R(t_k) = R_k
\end{cases}
\tag{5}
$$

We consider the general inverse problem in the following form: Given observational data $\{(t_i, y_i)\}_{i=1}^N$, where $y_i$ represents noisy measurements of one or more compartments of the SIR model, determine the unknown parameters $\beta$ and $\gamma$, along with the corresponding state trajectories $S(t), I(t), R(t)$.

Both complete data $(S(t), I(t), R(t))$ and partially observed data can be available for training, in particular, only infected $I(t)$, which is typical in epidemiological datasets.

The inverse problem is known to be ill-posed in the partial observation case due to strong correlations between the parameters $\beta$ and $\gamma$. In particular, different parameter combinations may yield nearly identical trajectories for $I(t)$, making reliable parameter recovery challenging. To ensure the stability of the solution, regularization is used, which effectively replaces the original ill-posed problem with a close-to-well-posed one. Within the framework of PINN, regularization is implemented through the inclusion of additional terms in the loss functional that reflect the physical constraints of the model, initial conditions, and a priori information about the parameters. This allows the solution to be selected from a set of admissible solutions as the one that best aligns with the model equations and observations. This approach aligns with the classical Tikhonov regularization theory, which states that a stable approximate solution to an ill-posed problem can be obtained by minimizing a functional that includes a penalty for deviations from the data and a regularization term. More information about inverse and ill-posed problems can be found in (Kabanikhin, 2018).

## 2.4 REDUCED PARAMETERIZATION VIA THE BASIC REPRODUCTION NUMBER

To mitigate identifiability issues, we also consider a reduced parameterization of the SIR model based on the basic reproduction number $\tilde{R}_0$. Introducing a rescaled time variable $\tau = \gamma t$, the SIR system can be rewritten as

$$
\begin{cases}
\frac{dS}{d\tau} = -\tilde{R}_0 S I, \\
\frac{dI}{dt} = \tilde{R}_0 S I - I, \\
\frac{dR}{dt} = I,
\end{cases}
\tag{6}
$$

In this formulation, the dynamics depend explicitly only on the single parameter $\tilde{R}_0$. This reduction decreases the dimensionality of the inverse problem and can significantly improve the stability of parameter estimation, particularly when observational data are limited.

The inverse problem in the reduced setting consists in identifying the value of $\hat{R}_0$ from the available data, while simultaneously reconstructing the state variables of the model.

# 3 REGULARIZED PINN FRAMEWORK

## 3.1 PINN APPROXIMATION OF THE SIR MODEL

Physics-Informed Neural Networks (PINNs) provide a flexible framework for solving forward and inverse problems governed by differential equations by embedding physical laws directly into the training process of neural networks. In contrast to purely data-driven approaches, PINNs leverage both observational data and the governing equations, which is especially advantageous in data-scarce regimes.

In this work, the state variables of the SIR model are approximated by a fully connected neural network

$$\mathcal{N}_\theta : t \rightarrow (\hat{S}(t), \hat{I}(t), \hat{R}(t)) \tag{7}$$

where $\theta$ denotes the set of trainable network parameters (weights and biases). The network takes time $t \in [0, T]$ as input and outputs the approximations of the three compartments.

Time and state variables are rescaled to comparable ranges from 0 to 1, reducing gradient stiffness and improving numerical stability during optimization.

## 3.2 LOSS FUNCTION CONSTRUCTION

A PINN is constructed to approximate the solution by minimizing a composite loss function combining residuals of the governing equations, penalties for violations of initial conditions, and mismatch terms for observational data.

PINNs often suffer from training instabilities, particularly in inverse settings where multiple competing loss terms must be balanced. Poor loss scaling may lead to non-physical solutions or inaccurate parameter recovery, even when observational errors are small (Wang et al., 2022). Separate weights are introduced for equation residuals, initial condition constraints, and data mismatch terms. This allows explicit control over the relative importance of physical consistency and data fidelity.

The total loss function takes the form

$$\mathcal{L} = w_{PDE}\mathcal{L}_{PDE} + w_{IC}\mathcal{L}_{IC} + w_{data}\mathcal{L}_{data}, \tag{8}$$

where $w_{PDE}, w_{IC}, w_{data}$ are weighting coefficients that balance the contributions of the individual terms.

The physics-based loss term enforces the SIR equations:

$$\mathcal{L}_{PDE} = \frac{1}{N_f} \sum_{j=1}^{N_f} \left( \left| \frac{d\hat{S}}{dt}(t_j) + \beta\hat{S}(t_j)\hat{I}(t_j) \right|^2 + \left| \frac{d\hat{I}}{dt}(t_j) - \beta\hat{S}(t_j)\hat{I}(t_j) + \gamma\hat{I}(t_j) \right|^2 + \tag{9}$$

$$+ \left| \frac{d\hat{R}}{dt}(t_j) - \gamma\hat{I}(t_j) \right|^2 \right) \tag{10}$$

where $\{t_j\}_{j=1}^{N_f}$ are collocation points sampled uniformly from the time interval.

The data mismatch loss penalizes deviations between model predictions and available observations:

$$\mathcal{L}_{data} = \frac{1}{N_d} \sum_{i=1}^{N_d} \left( ||\hat{y}(t_i) - y_i||^2 \right), \tag{11}$$

where $y_i$ denotes observed data for one or more compartments, depending on the considered scenario, $\{t_j\}_{j=1}^{N_d}$ are Collocation points for known data.

The initial condition loss enforces consistency with the prescribed initial state:

$$\mathcal{L}_{IC} = \left| \hat{S}(0) - S_0 \right|^2 + \left| \hat{I}(0) - I_0 \right|^2 + \left| \hat{R}(0) - R_0 \right|^2, \tag{12}$$

### 3.3 INVERSE PROBLEM FORMULATION IN THE PINN SETTING

In the inverse problem setting, the unknown epidemiological parameters are treated as additional trainable variables and optimized jointly with the neural network parameters. Two alternative formulations are considered.

In the first formulation, both transmission and recovery rates $\beta$ and $\gamma$ are assumed unknown. The loss function remains unchanged.

In the second formulation, the model is reparameterized using the basic reproduction number $\tilde{R}_0 = \frac{\beta}{\gamma}$, and the governing equations are written in the rescaled time variable. In this case, only $\tilde{R}_0$ is treated as an unknown parameter.

### 3.4 NETWORK ARCHITECTURE AND TRAINING STRATEGY

The neural network architecture consists of multiple fully connected layers with smooth activation functions. The number of hidden layers and neurons per layer is chosen to provide sufficient expressive power while avoiding overparameterization.

In the inverse problem setting, unknown model parameters are considered as separate "subnets" with separate outputs of the neural network. See figure 1.

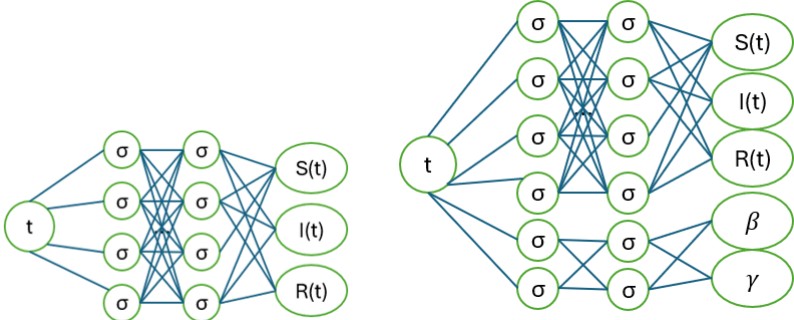

Figure 1: Left: PINN scheme for solving forward problem. Right: PINN scheme for solving inverse problem

Training is performed using gradient-based optimization with the Adam optimizer. To improve convergence and stability, the training process is divided into two stages. In the first stage, the network is trained with fixed parameter values to obtain a reasonable approximation of the solution. In the second stage, the epidemiological parameters are released and jointly optimized with the network weights.

The hyperbolic tangent activation function is used in this study. The neural network consists of four hidden layers, each of which contains 128 neurons. In the case of using a separate subnet to restore parameters, two hidden layers were used in it, while the main network retained a three-layer structure. The network was trained for 15000 epochs using a learning rate of $10^{-3}$.

### 3.5 HYPERPARAMETER OPTIMIZATION WITH OPTUNA

The performance of PINNs is highly sensitive to the choice of hyperparameters, particularly the weighting coefficients in the loss function.To systematically select these hyperparameters, we employ the Optuna framework for automated hyperparameter optimization in the inverse problem setting.

Each trial involved training the PINN model with a specific set of loss weights, and the objective function was defined based on the validation error of the parameters. Such an automated selection of loss weights helps to address the sensitivity of PINN training to manual tuning and improves

robustness with respect to noise and model stiffness. The weighting coefficients for the loss function were optimized using the Optuna library in the range of values from 0.1 to 100.

## 3.6 IMPLEMENTATION DETAILS

PINN is implemented in Python using the DeepXDE library, and the tensorflow framework was used as the backend. Automatic differentiation is used to compute all required derivatives. All experiments are conducted using the same training protocol to ensure fair comparison between different parameterizations and data availability scenarios. Synthetic datasets are generated by numerically solving the SIR model and adding controlled noise to simulate measurement uncertainty (13). The noise was determined by adding a normally distributed random variable multiplied by the data itself:

$$\begin{cases} S_{true} = S(t, \beta, \gamma) + \varepsilon S(t, \beta, \gamma) \\ I_{true} = I(t, \beta, \gamma) + \varepsilon I(t, \beta, \gamma) \\ R_{true} = R(t, \beta, \gamma) + \varepsilon R(t, \beta, \gamma) \end{cases} \quad (13)$$

where $\varepsilon \sim \mathcal{N}(0, 1)$.

## 4 NUMERICAL EXPERIMENTS

### 4.1 FORWARD PROBLEM: RECONSTRUCTION OF SIR DYNAMICS

We first consider the forward problem, where the epidemiological parameters are assumed known and the goal is to reconstruct the state trajectories $S(t), I(t), R(t)$.

The PINN accurately reproduces the reference numerical solution over the entire time interval. The reconstructed curves are smooth and remain consistent with the governing equations even in regions where observational data are sparse. If there is data for training only on I(t), PINN also shows fairly accurate results, see figure 2. Here solid lines are PINN predictions, points - reference solution with additive noise.

This experiment confirms that the proposed PINN architecture is capable of solving the direct SIR problem with high accuracy and provides a reliable foundation for the inverse problem.

### 4.2 INVERSE PROBLEM WITH TWO UNKNOWN PARAMETERS $(\beta, \gamma)$

We next consider the inverse problem in which both the transmission rate $\beta$ and the recovery rate $\gamma$ are treated as unknown and inferred from data.

When observations of all three compartments are available, the PINN successfully reconstructs both state trajectories and epidemiological parameters. The inferred values of $\beta$ and $\gamma$ closely match their true values, and the reconstructed trajectories overlap with the reference solution, see figure 3.

Predicted parameters are $\hat{\beta} = 0.809, \hat{\gamma} = 0.198$, when $\beta = 0.8, \gamma = 0.2$ for synthetic observation data.

This result demonstrates that the inverse problem is well-posed when sufficient observational information is available.

A more challenging scenario is considered where only the infected population $I(t)$ is observed. In this case, the PINN accurately reconstructs the $I(t)$ trajectory, while the susceptible and recovered compartments remain consistent with the dynamics implied by the equations.

However, parameter recovery exhibits a distinct behavior: the recovery rate $\gamma$ is identified accurately, whereas the transmission rate $\beta$ is not reliably reconstructed and exhibits large variability across runs. To select the optimal weights of the components of the loss function, a library for Python, Optuna, based on Bayesian optimization, was used. This optimization demonstrates that it is not possible to select weights in such a way that the error in determining the beta parameter is satisfactory. The Pareto front for this situation can be seen in figure 4.

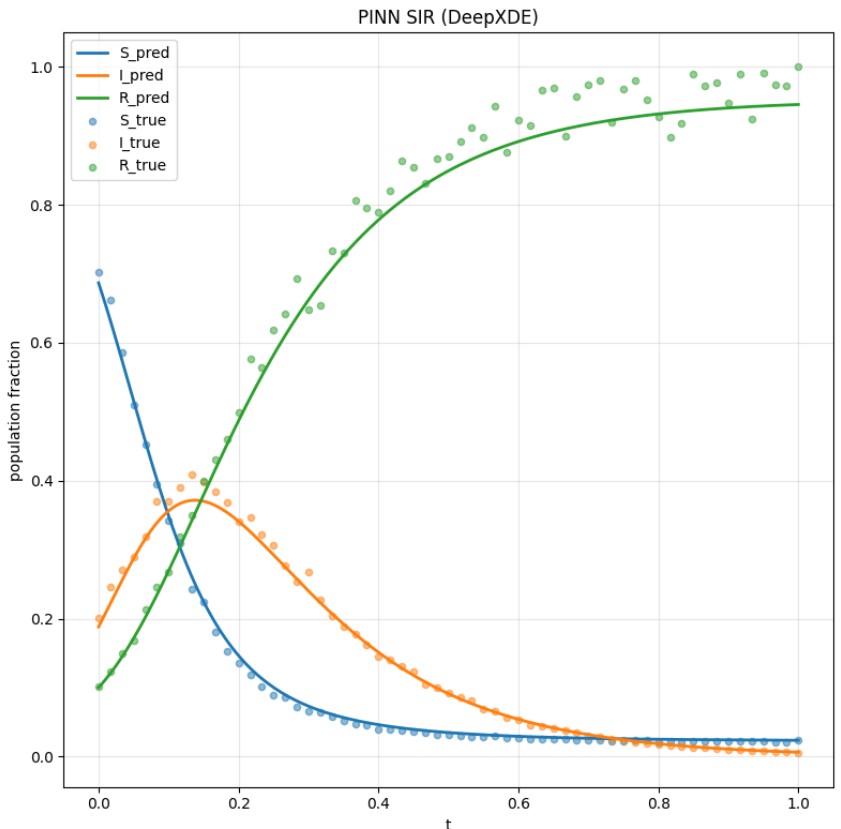

Figure 2: Comparison of reference SIR solution (dots) and PINN predictions (lines) for the forward problem (only data on I(t) for training).

| Weighting coefficient | Optimized value |
|---|---|
| PDE for the S variable | 4.294301104182 |
| PDE for the I variable | 3.2480544058569345 |
| PDE for the variable R | 8.215279160631946 |
| IC for the variable S | 0.9757002634952165 |
| IC for the variable I | 78.79919520702977 |
| IC for variable R | 4.445948626317913 |
| Data for variable I | 85.9842333854002 |

Table 1: Optimal parameters (optuna)

## 4.3 REDUCED PARAMETERIZATION VIA THE BASIC REPRODUCTION NUMBER $\tilde{R}_0$

To address the identifiability issues observed in the two-parameter formulation, the inverse problem is reformulated in terms of the basic reproduction number $\tilde{R}_0 = \frac{\beta}{\gamma}$. In this reduced formulation, only $\tilde{R}_0$ is treated as an unknown parameter, while the governing equations are rescaled accordingly. This significantly reduces the dimensionality of the inverse problem.

In this case, the following optimal parameters were determined:

: : : : : : :

The PINN successfully reconstructs both the state trajectories (see figure 5) and the value of $\tilde{R}_0$, even when only $I(t)$ observations are available. The dotted lines on the graph show the curves generated using the parameter recovered by optuna, $R_{optuna} = 4.0004$. Based on the comparison, it

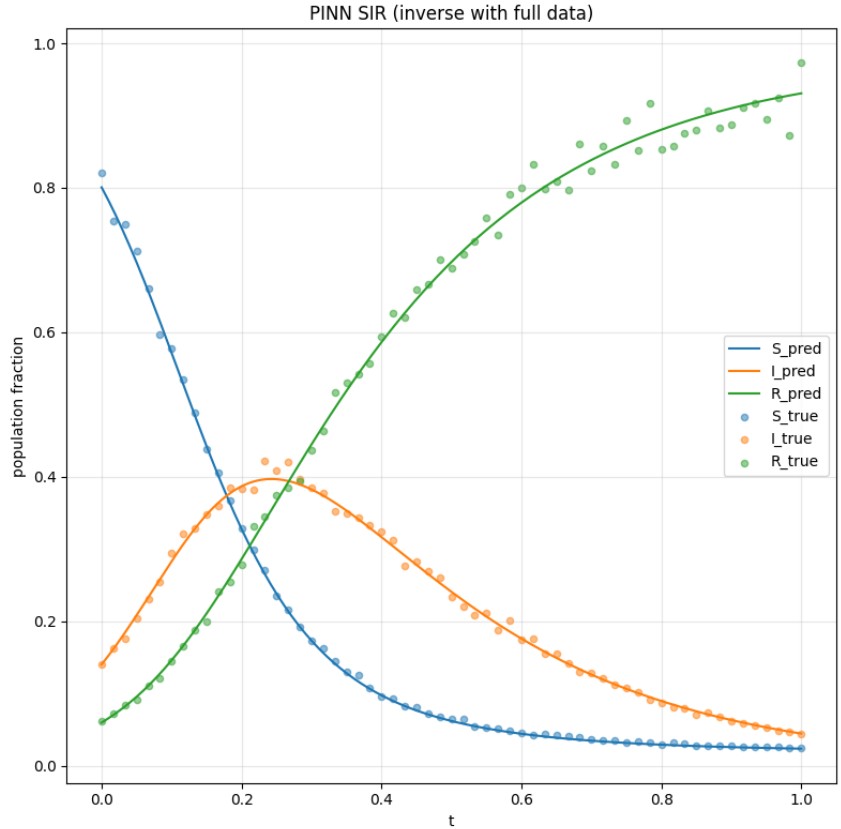

Figure 3: Reconstruction of $S, I, R$ (lines) using data for S(t), I(t), R(t) (dots).

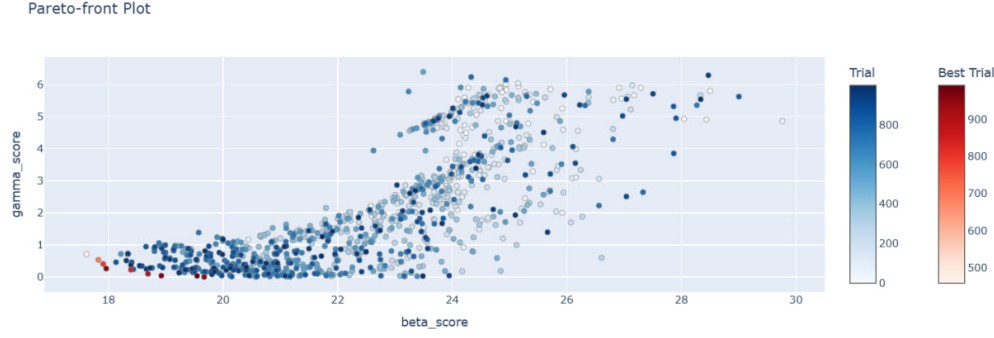

Figure 4: The Pareto front in the case of partial data

can be concluded that the curves are almost identical everywhere, even though there is a significant error in predicting the parameter using PINN.

Predicted parameter is $\hat{\tilde{R}}_0 = 3.874$, when $\tilde{R}_0 = 4.0$ for synthetic observation data. This result demonstrates that reparameterization improves robustness and parameter identifiability under limited data.

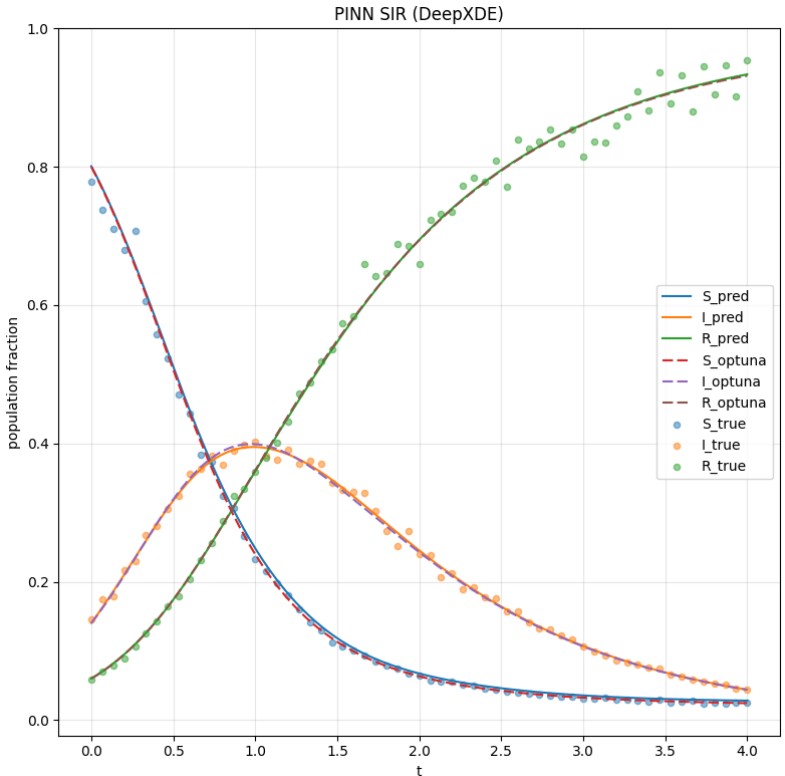

Figure 5: Reconstruction of $S, I, R$ by PINN (lines) using partial observations (only for I(t)), and reconstruction by optuna (dashed), reduced parameterization.

## 5    CONCLUSIONS AND FUTURE WORK

In this work, a physics-informed neural network framework for solving forward and inverse problems for the SIR epidemiological model has been developed and systematically analyzed. The proposed approach combines data-driven learning with explicit enforcement of the governing differential equations, allowing reliable reconstruction of both system dynamics and epidemiological parameters.

The numerical experiments demonstrate that PINNs accurately solve the forward SIR problem and provide stable reconstructions of state trajectories. For the inverse problem with two unknown parameters, the results show that full observations of all state variables enable accurate recovery of both the transmission and recovery rates. In contrast, when only partial observations are available, parameter identifiability becomes limited: the recovery rate can be inferred reliably, while the transmission rate cannot be uniquely reconstructed.

To address this limitation, the inverse problem was reformulated in terms of the basic reproduction number $\tilde{R}_0$. This reduced parameterization significantly improves robustness and allows accurate parameter recovery even under restricted observational data. The effectiveness of this reformulation is further supported by a global sensitivity analysis, which reveals pronounced differences in the influence of epidemiological parameters on the model output.

Future work will focus on extending the proposed framework to real epidemic data, where model parameters are often time-dependent and subject to external influences such as interventions and behavioral changes. An important direction is the incorporation of temporally varying transmission and recovery rates and the investigation of the capability of PINNs to identify such non-stationary parameters. In addition, the extension of the methodology to more complex compartmental models

and the integration of uncertainty quantification techniques represent promising avenues for further research.

## ACKNOWLEDGMENTS

This work was supported by the grant of the state program of the "Sirius" Federal Territory "Scientific and technological development of the "Sirius" Federal Territory" (Agreement № 26-03 date 07.07.2025).

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
