# OpenReview forum: "Regularized physics-informed neural networks for numerical solving of inverse problems in dynamical systems"
_mathai.club/MathAI/2026/Conference — 2026 Oral_

### Official Review · Reviewer_mQp7 · 2026-03-11
**Physically based regularized neural networks for numerical solution of inverse epidemic modeling problems based on the classical SIR model.**

**Rating:** 6
**Confidence:** 4

**Review:**

Summary.

This paper investigates the use of physics-informed neural networks (PINNs) for inverse problems in dynamical systems, with a focus on inverse epidemic modeling based on the classical SIR model. The authors aim to recover unknown parameters and hidden trajectories from full and partial observations. Their approach combines normalization of time and state variables, a weighted composite loss function including the differential-equation residual, initial conditions, and data mismatch, a two-stage optimization procedure, and a reduced parametrization based on the basic reproduction number, $R_0=\beta/\gamma$.

The main contribution is the demonstration that, under partial observations, inverse PINN identification for the classical SIR model becomes ill-conditioned. In particular, when only the infected compartment $I(t)$ is observed, separate estimation of $\beta$ and $\gamma$ becomes unstable, whereas parametrization through $R_0$ yields more robust results. The paper is therefore most valuable as a study of identifiability issues in inverse PINN formulations for epidemiological dynamical systems.

Strengths.

The paper addresses a relevant and well-motivated problem. Inverse problems in dynamical systems are often ill-posed, especially in epidemiological modeling, where observations are incomplete and model parameters may be strongly correlated. The topic is therefore important both mathematically and practically.

A major strength of the paper is its explicit focus on identifiability. Rather than presenting PINNs merely as flexible approximation tools, the authors emphasize whether the model parameters can actually be recovered in a stable and reliable way from the available data. This makes the work more substantial than a purely numerical study.

Another positive aspect is the clarity of the formulation. The loss function follows the standard PINN structure, the role of its components is easy to interpret, and the reduced parametrization via $R_0$ is both mathematically meaningful and practically motivated. The paper also deserves credit for honestly illustrating the limitations of the approach, particularly the fact that accurate trajectory reconstruction does not necessarily imply reliable parameter identification.

Weaknesses.

The main limitation of the paper is that the experimental validation relies entirely on synthetic data. While this allows controlled evaluation, it does not capture the complexity of real epidemiological observations, including reporting delays, under-reporting, model mismatch, and time-varying dynamics.

The proposed framework is better viewed as a stabilized inverse-PINN baseline than as a fundamentally new PINN methodology. The regularization effect arises mainly from loss weighting, normalization, and reparametrization, rather than from a new regularization principle or architectural innovation.

In addition, the paper provides insufficient detail for reproducibility. Important information about network architecture, activation functions, number of epochs, learning-rate schedule, hyperparameter search ranges, and final loss weights is missing. The work is also only weakly positioned relative to the broader literature, including( for example) self-adaptive PINNs, Bayesian PINNs, split-training strategies, observability-aware methods, and denoising-based approaches.

Recommendation.

Overall, this is a relevant and reasonably well-formulated study of inverse PINNs for the classical SIR model. Its main strength is the clear demonstration that partial observations fundamentally change the nature of the inverse problem and that reparametrization can act as a practical stabilization mechanism. In this sense, the paper is useful both as a baseline contribution and as an illustration of the identifiability problem in inverse epidemic modeling.

The paper would have stronger scientific impact if several aspects were improved. First, the literature review should be broadened, especially with respect to prior work on inverse PINN models. Second, the authors should explain more clearly how their formulation differs from earlier inverse-PINN approaches and what the specific methodological contribution of the paper is. Third, the role of Tikhonov regularization should be clarified, since the composite PINN loss already plays a regularizing role. Fourth, it would be valuable to compare reparametrization-based regularization with alternative strategies such as weighted-loss, split/architecture-based, identifiability-aware, and denoising-based regularization. Finally, the paper should provide a more complete description of the network architecture and training procedure in order to improve reproducibility.

In summary, the paper is methodologically useful and addresses an important issue in inverse PINN modeling for the classical SIR system, but it would benefit from a broader literature context, clearer positioning of novelty, and a more detailed computational description.

---

> ### Author Rebuttal · Authors · 2026-03-13
>
> - "Experimental validation relies entirely on synthetic data"
> Answer: In the process of further work, it is expected to carry out validation on real data. To do this, it is necessary to complicate the model by introducing a time dependence of parameters or adding additional equations to the system.
> The proposed framework is better viewed as a stabilized inverse-PINN baseline than as a fundamentally new PINN methodology.
> Answer: In this study, we use a previously developed PINN model instead of a new one. The goal of this study is to evaluate the applicability of this model to ill-posed inverse problems.
>
> 	-  "Important information about network architecture, activation functions, number of epochs, learning-rate schedule, hyperparameter search ranges, and final loss weights is missing."
> Answer: The hyperbolic tangent activation function is used in this study. The neural network consists of three hidden layers, each of which contains 64 neurons. In the case of using a separate subnet to restore parameters, two hidden layers were used in it, while the main network retained a three-layer structure. The network was trained for 10,000 epochs using a learning rate of 10^-3. The weighting coefficients for the loss function were optimized using the Optuna library in the range of values from 0.1 to 100. As a result, the following optimal parameters were determined:
> Weighting coefficient for the partial differential equation (PDE) for the S variable: 4.294301104182
> Weighting coefficient for the partial differential equation (PDE) for the I variable: 3.2480544058569345
> Weighting factor for the partial differential equation (PDE) for the variable R: 8.215279160631946
> Weighting factor for the initial conditions (IC) for the variable S: 0.9757002634952165
> Weighting factor for the initial conditions (IC) for the variable I: 78.79919520702977
> Weight coefficient for initial conditions (IC) for variable R: 4.445948626317913
> Weight coefficient for data for variable I: 85.9842333854002
> These values will be added to the text of the article.
>
> 	- "The work is also only weakly positioned relative to the broader literature"
> Answer: As part of the scientific literature review on Physics-Informed Neural Networks (PINN), the proposed modifications of this methodology were thoroughly examined. Nevertheless, at the initial stage of the research, a more conventional PINN architecture was selected.

---

### Official Review · Reviewer_GtXv · 2026-03-12
**the article represents a serious study in the field of applying neural network methods to solve inverse problems. The work is characterized by a high scientific level, relevance and practical significance. It is recommended for publication after revision taking into account the specified remarks.**

**Rating:** 7
**Confidence:** 4

**Review:**

The presented article is devoted to the actual problem of applying neural network methods to solve inverse problems in dynamic systems using the example of epidemiological models. The work is located at the intersection of mathematical modeling, machine learning and epidemiology.
The scientific significance of the work is determined by the growing role of artificial intelligence methods in solving complex mathematical problems. The use of PINN (Physics-Informed Neural Networks) to solve inverse problems in epidemiological models is a promising area of research.
The main goal of the work is to develop and investigate a regularized approach based on PINN for solving inverse problems in epidemiological models. The objectives of the research are clearly formulated and correspond to the stated goal.
Main achievements of the work is development of a new approach to solving inverse problems using regularization and application of time and phase variable normalization.
The practical value of the work lies in the development of a method that can be applied for reconstruction of hidden state trajectories, iIdentification of epidemiological parameters and processing incomplete and noisy data.

Remarks to the work:
- Insufficient attention to the analysis of computational complexity of the method;
- Lack of comparison with alternative approaches;
- Limitation of experiments to synthetic data.

As a conclusion, I would like to say that the article represents a serious study in the field of applying neural network methods to solve inverse problems. The work is characterized by a high scientific level, relevance and practical significance. It is recommended for publication after revision taking into account the specified remarks.

---

> ### Author Rebuttal · Authors · 2026-03-13
>
> - "Insufficient attention to the analysis of computational complexity of the method"
>  Answer: During the study, various configurations of the neural network architecture were analyzed, including the number of hidden layers and neurons, as well as the number of training epochs. It was found that as the network parameters or the number of epochs increase beyond a certain threshold, the increase in accuracy becomes insignificant. These dependencies will be presented in the form of additional graphs.
>
> - "Lack of comparison with alternative approaches"
> Answer: This study did not involve a comparative analysis of different methods for solving the inverse problem using PINN. However, it may be useful to consider comparing the results with those obtained using the Optuna platform. This comparison will be presented on an additional graphs.
>
> - "Limitation of experiments to synthetic data"
> Answer: In the process of further work, it is expected to carry out validation on real data. To do this, it is necessary to complicate the model by introducing a time dependence of parameters or adding additional equations to the system.

---

### Official Review · Reviewer_vPiZ · 2026-03-13
**Regularized Physics-Informed Neural Networks for Numerical Solving of Inverse Problems in Dynamical Systems**

**Rating:** 7
**Confidence:** 3

**Review:**

The article "Regularized Physics-Informed Neural Networks for Numerical Solving of Inverse Problems in Dynamical Systems" explores the application of physically-informed neural networks to solving inverse epidemiological modeling problems using the SIR model as an example. The authors propose a regularized approach with variable normalization, weighting of loss function components, and automatic hyperparameter selection using Optuna. The work is well-executed, contains a clear mathematical formulation, and a thorough analysis of the identifiability of the parameters under limited data conditions. It is shown that when observing only infected individuals, the traditional formulation does not allow for reliable recovery of both parameters; however, reparametrization using the basic reproduction number significantly improves the result. The main criticisms relate to limited novelty (the approach is already known) and the lack of validation on real data; however, this does not detract from the overall positive impression. The article is relevant to the theme of the "Mathematics and Artificial Intelligence" conference and is recommended for acceptance.

---

> ### Author Rebuttal · Authors · 2026-03-13
>
> -"Limited novelty (the approach is already known)"
> Answer: As part of further research, it is planned to complicate the analyzed system by adding additional unknowns and differential equations. Due to the solution of similar and more complex problems, the use of traditional numerical methods, such as the finite difference method, may not provide the necessary efficiency. Therefore, it seems appropriate to consider alternative methods for solving such problems. Although the use of physical invariant neural networks (PINN) is not a fundamentally new approach, this work focuses on their application to ill-posed inverse problems.
>
> - "Lack of validation on real data"
> Answer: In the process of further work, it is expected to carry out validation on real data. To do this, it is necessary to complicate the model by introducing a time dependence of parameters or adding additional equations to the system.

---

### Decision · Program_Chairs · 2026-03-14

**Decision:**

Accept (Oral)

**Comment:**

Dear Author(s),

On behalf of the Program Committee of the International Conference on Mathematics of Artificial Intelligence (MathAI 2026), we are pleased to inform you that your paper has been accepted for an oral presentation at MathAI 2026.

Your paper was evaluated through a rigorous two-stage review process involving both automated screening and expert review by members of the Program Committee. The reviewers recognized the quality and contribution of your work.

Presentation details:

- Format: Oral presentation (15–20 minutes + 5 minutes Q&A)
- Mode: You may present either in person (offline) at the conference venue in Sirius, Russia, or remotely via Zoom. Please indicate your preferred mode when confirming your participation.
- Conference dates: Marh 30 - April 3, 2026
- Website: https://mathai.club

Next steps:

1. Please confirm your participation and presentation mode by replying to this email mathai.club@yandex.ru no later than March 15, 2026 18:00 Moscow time.
2. If you plan to attend in person, the organizing committee will provide accommodation details separately.
3. Please prepare your final camera-ready manuscript according to the formatting guidelines available at https://mathai.club and upload it to OpenReview by March 15, 2026 18:00 Moscow time.

Should you have any questions regarding the program, logistics, or your presentation slot, please do not hesitate to contact us.

We look forward to your contribution to MathAI 2026.

With kind regards,

MathAI 2026 Program Committee
International Conference on Mathematics of Artificial Intelligence
https://mathai.club
OpenReview: https://openreview.net/group?id=mathai.club/MathAI/2026/Conference
Telegram: https://t.me/MathAI_club
Email: mathai.club@yandex.ru